# Solar UV Measured under Built-Shade in Public Parks: Findings from a Randomized Trial in Denver and Melbourne

**DOI:** 10.3390/ijerph191710583

**Published:** 2022-08-25

**Authors:** Suzanne Jane Dobbinson, David Bard Buller, James Andrew Chamberlain, Jody Simmons, Mary Klein Buller

**Affiliations:** 1Centre for Behavioural Research in Cancer, Cancer Council Victoria, Melbourne, VIC 3004, Australia; 2Klein Buendel, Inc., Golden, CO 80401, USA; 3Cancer Epidemiology Division, Cancer Council Victoria, Melbourne, VIC 3004, Australia; 4Office of the Deputy Vice-Chancellor, LaTrobe University, Melbourne, VIC 3086, Australia

**Keywords:** landscape change, parks/trails, greenspace, built environment, ultraviolet radiation, shade, public health, skin cancer prevention

## Abstract

Reducing exposure to ultraviolet radiation (UV) is crucial for preventing UV-induced diseases of the skin and eyes. Shade may offer significant protection from UV. More empirical research to quantify the UV protection offered from built shade is needed to guide disease prevention practices and confidence in investment in shade. This study quantified UV levels under built-shade relative to unshaded passive recreation areas (PRAs) over summer months in parks in two cities. In a randomized controlled trial, *n* = 1144 UV measurements were conducted at the center and periphery of PRAs in a total sample of 144 public parks as part of pretest and posttest measures of use of the PRAs by park visitors for three recruitment waves per city during 2010 to 2014. Following pretest, 36 PRAs received built-shade and 108 did not. Regression analyses modelled pre-post change in UV (Standard Erythemal Dose (SED) per 30 min) at PRAs; and environmental predictors. Mean UV at the center of built-shade PRAs decreased from pretest to posttest ($\bar{x}$ = 3.39, $\bar{x}$ = 0.93 SED), a change of $\bar{x}$ = −3.47 SED relative to control PRAs (*p* < 0.001) adjusting for the covariates of ambient SED, (cosine) solar elevation and cloud cover. Clouds decreased and solar elevation increased UV levels under shade. No significant differences in UV by shade design occurred. A substantial reduction in exposure to UV can be achieved using built-shade with shade cloth designs, offering considerable protection for shade users. Supplementary sun protection is recommended for extended periods of shade use during clear sky days.

## 1. Introduction

The built-shade industry has steadily grown in hot climates around the world [1]. Shade can be an attractive feature with creative designs used to energize public open space. Importantly, shade offers protection from climate extremes and from ultraviolet radiation (UV), thus increasing the usability and amenity of outdoor areas [2,3]. UV is an integral component of sunlight reaching the Earth’s surface. Ozone and clouds in the upper atmosphere filter out nearly all the harmful high-energy radiation from the sun [4]. Nonetheless, a very small amount of biologically active UVA and UVB radiation reach the Earth’s surface; 6.5% and 0.04% of total solar irradiance, respectively [4,5]. Ambient temperatures and UV may reach extreme levels during summer months in many regions, including in Denver, Colorado, United States (USA) and Melbourne, Victoria, Australia, when the sun is at its seasonally highest for the locality, with peak UV intensity at solar noon during summer solstice [4]. Regions at higher elevation also have increased UV levels, with up to a 10% increase in UV per 1000 m above sea level [6].

Exposure to UV is damaging to skin and eyes [7,8,9,10,11]. Melanoma causes the most mortality of the UV-induced diseases [9] and is largely preventable with reduction in exposure to UV. It is estimated that in 2012, 75.7% of new cases of melanoma worldwide were attributable to excess UV exposure [12]. Melanoma is among the most common cancers in light-skinned populations, with age-standardized rates (ASR) highest in Australia (ASR 36.6), New Zealand (ASR 31.6), Denmark (ASR 29.7), other countries in North West Europe, USA (ASR 16.6) and the United Kingdom (ASR 16.0) [13]. In the USA alone, 106,110 new cases of melanoma were estimated to be diagnosed in 2021 [14]. Keratinocyte cancers of the skin (basal and squamous cell carcinomas) have a much higher prevalence than melanoma in both Australia and the USA and are also caused by UV exposure [7,14]. Skin cancer prevention advocates recommend the use of shade and other forms of sun protection to reduce exposure to UV and skin cancer risk [15,16]. Shade development is actively promoted in a range of settings through policy development [15,16].

Parks can contribute to the physical, psychological, and social health of communities [17,18]. Shade is considered a desirable feature by individuals who use public parks [19,20] and health authorities worldwide advise that individuals should seek shade when outdoors to reduce their exposure to solar UV [21,22]. Natural shade from trees is ideal for parks, providing both UV protection [23] and cooling [24,25,26,27], as well as added benefits of trapping carbon dioxide, producing oxygen during photosynthesis [28], and providing habitat for wildlife [29]. However, tree shade is not universally available in parks, as the result of, for example, development of new suburbs and droughts constraining tree plantings. Purpose-built shade may be appropriate in these situations to ensure that the cooling benefits and protection from UV and glare from intense sunlight are available to park visitors ubiquitously. Research from temperate climates suggests that type of shade is an important determinant of use. To be most desirable and effective, purpose-built shade should be designed to: (i) create light-colored shade of a relatively large size [30] and; (ii) provide shade during the hours close to solar noon in summer when UV is at its peak. Our recent research utilized these principles to experimentally test the effect of adding purpose-built shade to areas in parks on usage of these newly shaded areas. The study outcomes suggested that such built-shade using tensioned shade cloth may increase the number of individuals shaded in parks [31].

There is currently limited research on the UV protection provided by different types of built-shade in situ [32]. Most of these studies indicate highly variable levels of UV protection offered from shade cloth structures [32,33]. Manufacturers’ technical ratings of the Ultraviolet Protection Factor (UPF) of shade cloth products give some assurance that a selected product will sufficiently block UV. However, the effectiveness of the UV protection from a given shade structure in situ will vary dependent on many factors [32]. The current research aims to contribute to a better understanding of the factors that affect UV protection of built-shade structures. Additionally, the extent to which individuals who use the shade will be protected from sunburn, and for how long, can be derived from accurate UV measurements under shade in situ. This is important given that sunburn experience is associated with increased melanoma risk [34]. This information will be useful to better educate the public about the benefits and limitations of shade for skin cancer prevention.

## 2. Materials and Methods

Solar UV was measured under purpose-built shade sails at passive recreation areas (PRAs) in public parks during summer months in cities located in two hemispheres, namely, Denver, Colorado, USA, 39.7392° N, 104.9903° W, and Melbourne, Victoria, Australia, 37.8136° S, 144.9631° E. The UV measures were conducted as part of a randomized trial of built-shade during 2010 to 2014, using a pretest-posttest controlled design (Figure 1). The primary outcomes of the trial were previously described and examined the use of this shade by park visitors [31]. Solar UV was measured as a secondary outcome in the trial. Analyses compared (i) changes in UV exposure at the study PRAs before and after purpose-built shade was constructed and (ii) changes between PRAs where shade was built and those that remained unshaded. We hypothesized that the introduction of shade would decrease the average UV exposure in the PRA [35]. A secondary aim was to examine environmental predictors (including shade-design features) of the measured UV under the shade.

### 2.1. Enrolment of Passive Recreation Areas

A total of *n* = 144 study PRAs were enrolled in three annual waves in each city prior to pretest. City and municipal parks staff provided listings of public parks, excluding parks likely to be ineligible, for example, due to few target amenities or to scheduled park refurbishments. Subsequently, each park was assessed by research staff via onsite audits and reviews of photographed areas to identify those with PRAs suitable for the study. To be eligible, PRAs had to: (i) be located in public parks containing at least two unshaded PRAs; (ii) meet the definition of a PRA; and (iii) be in full sun between 11 am and 3 pm during summer months; (iv) contain a space where a shade sail could be constructed and; (v) be approved by parks department staff for shade sail construction. One of the two PRAs in each park was selected for full assessment, including measurements of UV, and potential randomization to shade construction. The alternate study PRA was simply observed to estimate the park’s overall use by park visitors. Thus, only one main study PRA was used for assessment of primary outcomes in each park to avoid bias due to clustering of PRAs within a park.

### 2.2. Procedures

For each study wave, pretest measures commenced in the first summer following PRA enrolment. The parks were subsequently randomized by an independent biostatistician allocating treatment, to a shade sail construction, and control parks, to remain unshaded, on an unequal 1 to 3 ratio by city-wave stratum using the random number generator procedure in Stata. Shade construction at treatment PRAs was completed prior to posttest measures. Posttest measures commenced in the second summer following enrolment for the wave. In pretest and posttest on four weekend dates per park, trained observers recorded the use of PRAs during 30-min periods, followed by measurement of the UV levels at the center and boundary (or periphery) of the PRAs. Summer assessments were scheduled between 11 am and 3 pm DST over four months (Denver: June to September; Melbourne: December to March); on alternating Saturdays and Sundays, and early and late in the four-hour period. Observation occurred when the forecast high temperature was between 72° F (22 °C) and 95° F (35 °C) and suspended during rain. The project biostatistician was blind to the condition. However, the study condition was apparent to data collection staff as it was impossible to conceal the shade sails.

#### 2.2.1. Treatment: Built-Shade

Shade structures were built to similar designs in both cities. The designs varied to fit the site requirements and the preferences of city and municipality parks department staff. The designs were also intended to create attractive shade structures, to maximize available shade from 11 am to 3 pm DST in summer, and to comply with all local engineering, building, and planning codes. Shade cloth with a minimum UPF rating of 94% reduction in UV was used for the structures. A variety of colors of shade cloth were used, which provided variable light transmission. We defined shade cloth colors as ‘light’, transmitting 19–25% visible light, or ‘dark’, transmitting less than 14% visible light. Project staff recommended that the shade be the largest size acceptable to the city and municipality parks department staff. Entry pole heights and shade cloth area were obtained from the construction plans. The designs built were tensioned shade sails, hip and ridge structures with heavier frames, cantilever structures with only two support poles utilized to free pathways and playground areas of poles, and a custom wrap shade design. Only shade-sails were built in Melbourne PRAs. In Denver, shade designs were more variable to account for snow load and city staff preferences. Refer to Figure 2a–h for photographs of a selection of the structures built. Ownership of the shade sails was transferred to the City and municipalities once built to compensate them for work on the project and allow for their continued upkeep.

#### 2.2.2. Solar UV Measurements

At the end of each observation period the research staff recorded the solar UV levels at the boundary and center of the PRA. The solar UV was measured using handheld meters, the Solarmeter UVR 6.5 SiC, which displayed the solar UV as an instantaneous reading of the ultraviolet index (UVI). The UV meters, Solarmeters, were purchased in 2010 from Solar Light Company, LLC located in Glenside, PA, USA and had a spectral responsivity based on 280–400 nm Diffey Erythemal Action Spectrum. The UV meters were calibrated by the manufacturer. Research staff were trained to hold the meter at arm’s length in a vertical position pointing the sensor directly up until a stable reading was determined. Research staff manually recorded two readings of the UVI for each location, with the average deemed the final measure in analyses. A total of 1152 UVI measurements, *n* = 576 per test period, for each position were recorded during the study. However, UV measures for *n* = 8 PRA observations were excluded from the analyses due to missing data for cloud measures. Use of a tripod may have improved the consistency of our UV measures. However, the meter was small and designed to be used by hand and we needed the research staff to be unobtrusive in the parks so as not to unduly influence our primary outcome measure for the trial, i.e., use of the shaded PRAs. Pilot testing determined that measurements from the handheld meter had good concordance with scientific instruments measuring UV (i.e., spectroradiometer and Biometer), good agreement with an electronic dosimeter, and good replication of measurements across different research staff, although small aberrations due to position occurred [36]. Two other studies reported these handheld meters underestimated absolute UVI levels by 10–20% [37], and varied by 4–8% with both over- and under-estimates [38].

#### 2.2.3. Observations of Environmental Conditions

The research staff recorded environmental conditions at each park site at the close of observations. There is considerable evidence that ambient UV levels decrease with increasing cloud cover and increase with increasing solar elevation as a function of days from summer solstice, hours from solar noon, and location [4]. Cloud cover was recorded as none, thin, thick, or overcast, and the percentage of the sky covered by clouds was also recorded. A cloud fraction (0% to 100%) was derived from these measures [35]. A subjective rating of wind speed on a 5-point scale (none to very strong) was also recorded; an anemometer was not used to avoid calling attention to the project staff. Temperatures for each observation period were obtained from city meteorological stations by project staff. The solar elevation angle, a measure of the sun’s angle for a particular location/time/date, was determined for the date and finishing time of each PRA observation and approximate location (latitude and longitude for each city), using the National Oceanic and Atmospheric Administration (NOAA) solar calculator [39].

### 2.3. Statistical Analysis

A standard erythemal dose (SED) is recognized as a useful measure of biologically-effective UV [4]. Thus, we defined UV outcomes for UV PRA measurements in the study as the exposure for a potential PRA user during a 30-min observation period expressed in SED (J/m^2^), using the formula SED (J/m^2^) = (UVI units measured by the solarmeters at the PRA × 0.025 W/m^2^ × number of seconds exposure)/100. However, we acknowledge that our measure of SED is less precise than SED measured using the CIE Erythema Action Spectrum, given that the Solarmeters measured UVI weighted to the Diffey Erythemal Action Spectrum [40]. Additionally, we transformed the SED into a minimal erythemal dose (MED) to aid interpretation of the reduction in sunburn risk under shaded PRAs [7]. One SED unit is 100 J/m^2^, with a dose of 2 SED equivalent to 1 minimal erythemal dose (MED) and sufficient to cause erythema for Fitzpatrick Skin Type I [4,7]. Notably, an individual with Fitzpatrick Skin Type I has skin that is most sensitive to UV [41] and fair skin types collectively have a high risk of skin cancer [34].

The efficacy of the shade structures in blocking the ambient UV was assessed by predicting the mean SED at the center of treatment PRAs relative to the mean SED at control PRAs during pretest compared with posttest. A linear mixed model (Stata mixed effects regression) was used to predict PRA center SED by experimental group (control/treatment). Study wave, city, test-period, cloud cover, the solar elevation angle during the observation time/location and the PRA boundary SED were entered in the model as fixed effects. The non-independence of SED measurements within parks at baseline was accounted for by entering park ID as a random effect using a single level random intercept model. A cosine transformation was used for solar elevation to convert values from degrees to 0–1 to aid interpretation. The cosine of the solar elevation angle was entered as a linear term with interactions with test period, cloud cover, experimental group, and group by test period. Additionally, a quadratic term interacted with cloud cover was included. Robust standard errors, calculated by Huber–White sandwich estimators in Stata, were used in the models when calculating confidence intervals. The use of robust standard errors gives valid inference in the situation where the residuals are independent of the covariates, but not necessarily identically distributed, with some evidence of this for solar elevation in the current analyses. Sample bias was minimized by random assignment of parks into treatment or control group. Additionally, SED measurements were only conducted at one PRA per park to reduce bias from PRA clusters. There was potential bias from multiple measurements of SED conducted at each PRA. However, the model of center PRA SED showed variance associated with PRA was small (0.0372). Therefore, to simplify post-estimation of predictive values, the results of the linear mixed model were regarded as preliminary. The final model of predictors of center PRA SED used multiple linear regression, in which the repeated SED observations for individual PRAs were considered independent observations. To refine model terms, the homoscedasticity of residuals and the significance of the regression coefficients were examined. Additionally, SED at the boundary of the PRAs was modelled using multiple linear regression to assess whether change in ambient SED explained change in SED at the PRA center by experimental group and test-period. The predicted values of these regression models were used to graph the estimated center and boundary SED for specific study conditions and test-periods in Denver and Melbourne separately.

Subsequently, we examined the relationship between shade design factors and UV levels under shaded PRAs at posttest using a series of linear mixed models within the constraints of non-randomized assignment of these features. We hypothesized a priori that: (i) PRAs with a hip and ridge design would have lower SED compared with shade sails and cantilever designs given this design-type offers greater protection against scattered UV; (ii) PRA designs with a larger shade cloth area would have lower SED than smaller shade cloth designs (cloth area in m^2^, shade radius in degrees), and; (iii) PRA SED would increase with height due to the sun being able to reach further under the shade for a given solar elevation angle. A standardized variable √A was used to assess effects of cloth size in these analyses following review of distributions and initial exploratory models. Linear mixed models were appropriate for these analyses given a higher variance in the PRA effect for shaded PRAs at posttest. The shade design, cloud and solar elevation variables were entered as fixed effects and park ID as a single random effect. SAS PROC MIXED with Restricted Maximum Likelihood (REML) was used to fit the models.

## 3. Results

### 3.1. Sample of SED Measurements

A total of 1144 SED measurements were calculated for the PRA center and the PRA boundary from the UVI measurements at each test-period. Table 1 describes the cloud cover, days from summer solstice and time to solar noon during the observations. There was considerable variation in cloud cover by city, with observations in Denver mostly conducted during clear-sky conditions compared with a high frequency of observations during thin-cloud and overcast conditions in Melbourne, likely reflecting climatological differences between these cities (e.g., less humidity in Denver than Melbourne). Solar elevation angles varied for each PRA SED observation due to time of day and days from summer solstice.

### 3.2. UV Exposure at PRAs during 30 Min Potential Usage

In this sub-section we present the unadjusted and predicted mean UV of shaded versus unshaded PRA conditions by test-period. Table 2 shows the unadjusted mean UV levels were lowest at the center of shaded PRAs during posttest compared with the PRAs in unshaded conditions. For further details of unadjusted mean UV levels refer to Appendix A, Figure A1a,b. A linear mixed model of center PRA SED adjusting for ambient SED, (cosine) solar elevation and cloud cover (refer to Appendix A, Table A1, Table A2 and Table A3) confirmed that the reduction in mean SED at shaded PRAs during posttest was statistically significantly lower than the other three experimental/test-period conditions (−3.47, 95% CI −4.65 to −2.30, P (z) < 0.001). This model also found that the UV levels at the center of the PRAs were significantly associated with the boundary (proxy ambient) SED (coefficient 0.86, 95% CI 0.83 to 0.91, P (z) < 0.001). Solar elevation angle was associated with PRA SED when interacted with the study condition and test-period, which significantly moderated the reduction in SED provided by shaded PRAs at posttest (coefficient 2.89, 95% CI 0.88 to 4.89, P (z) = 0.005).

### 3.3. Confirmatory Linear Regression—UV of Shaded PRA versus Unshaded PRA Conditions by Test-Period with Environmental Predictors

The linear regression model of center PRA SED (refer to Appendix A, Table A4) confirmed the findings of the linear mixed model, with a statistically significant reduction in SED under shaded PRAs at posttest relative to other study group/test-period conditions estimating that SED was approximately four units lower, on average, than the other conditions, holding the other predictor variables constant (coefficient −4.29 SED, 95% CI −5.69 to −2.89, P (t) < 0.001). In this regression model, which excluded ambient SED measures (PRA boundary), the presence of clouds was an additional significant predictor of center PRA SED with lower mean SED when clouds were present (cloud coefficient: −3.74 SED, 95% CI −5.60 to −1.87, P (t) < 0.001). Clouds also moderated the effect of cosine elevation with SED (cloud × solar elevation^2^ coefficient: −11.38 SED, 95% CI −16.89 to −5.86, P (t) < 0.001).

### 3.4. UV of Shaded PRA Center versus PRA Boundary by Test-Period with Environmental Predictors

The proxy ambient measures of UV at the PRAs showed no evidence of a change in UV levels from pretest to posttest at shaded PRAs relative to control PRAs. Specifically, a linear regression model of boundary PRA SED during pretest and posttest (refer to Appendix A, Table A5) found that there was no significant difference in boundary PRA SED from pretest to posttest by experimental group (shaded posttest coefficient: −0.78 SED, 95% CI −2.32 to −0.76 (P (t) = 0.321). The detailed results of parameters from the linear mixed model and these two linear regression models are presented in Appendix A, Table A1, Table A2, Table A3, Table A4, Table A5 and Table A6. The post-estimation predicted values from these linear regression models were used to describe the SED for specific environmental conditions during observations at PRAs in each city in the figures below.

Figure 3a,b show the change from pretest to posttest in center PRA and boundary PRA SED during clear sky observations for each experimental group by solar elevation angle in each city separately. The figures show the substantial reduction in UV from pretest to posttest at the shaded PRAs in both cities. For example, in both cities during observations with clear skies when solar elevation angles were high (~75°), the mean predicted UV levels at all PRAs approached five SED during pretest observations with a reduction to below two SED under shaded PRAs during posttest, while UV in the control PRAs and at the boundary of treatment PRAs remained high during posttest. Although figures for Denver and Melbourne are presented separately due to different solar elevation calculations for each hemisphere, the pattern of PRA SED in each city was similar. Nonetheless, when solar elevation angles were low, PRA SED were slightly higher in Denver compared with in Melbourne. For a solar elevation angle of 45°, the mean UV values at pretest were ~2.5 SED in Denver and ~2.2 SED in Melbourne, reducing to mean values below one for shaded PRAs in both cities during posttest.

### 3.5. UV Exposure by Cloudy Skies and Solar Angle

Figure 3a,b additionally depict the generally positive linear relationship between PRA SED and solar elevation. However, the mixed effects and regression models suggest this relationship is somewhat complex, with the effect of solar elevation angle on center PRA SED moderated by cloud cover (refer Table A4) including both linear (linear regression coefficient for cloud presence: 9.95 SED, 95% CI 2.27 to 17.63, *p* = 0.011) and curvilinear components (linear regression coefficient for cloud presence: −11.38 SED, 95% CI −16.89 to −5.86, *p* < 0.001).

Figure 4a,b describe the UV levels (SED) under shaded PRAs and at the boundary of these PRAs (proxy ambient) within each city during posttest by cloud cover and solar elevation angle. The figures overlay the plots with vertical lines to indicate the predicted SED for specific morning and afternoon observation times and solar noon during one mid-summer date and one end-of-summer date. Although the overall pattern of SED at the center and boundary of shaded PRAs for clear skies during posttest is the same as presented in Figure 3a,b, these figures highlight the variation in SED at shaded PRAs by cloud cover and its interaction with the effect of the solar elevation angle on PRA SED. The plots depict the linear increase in shaded PRA SED with increasing solar elevation angle during clear skies, while a nonlinear pattern of increase is observed in the presence of cloud when PRA SED were significantly lower.

Under cloudy skies, the SED at center and boundary PRAs in both cities steadily increased with solar elevation up until 60° and subsequently decreased (center PRAs) or rate of increase slowed (boundary PRAs) with solar elevation (refer Figure 4a,b). Additionally, the UV levels under shaded PRAs were further attenuated at solar elevation angles above 60° under clouds and the reduction in center PRA SED was increased relative to the boundary PRA (proxy ambient). Moreover, under cloudy skies, the UV levels at the shaded PRA did not reach above one SED, relative to one SED to approximately two SED during clear skies, for any observed solar elevation angle in either city. However, regardless of environmental conditions the average UV levels at the center of the shaded PRA did not exceed two SED during any 30-min exposure period in either city.

In both cities, during clear skies, PRA SED was highest at solar noon on summer solstice when the sun was at its zenith position, namely on June 21 in Denver and December 21 in Melbourne. For a given observation day, the amount of time to solar noon was correlated with solar elevation, resulting in lower PRA SED at times other than solar noon. This varied by city, for example, the solar elevation angles of PRA observations at 11 am in Melbourne on summer solstice were lower than at 11 am in Denver on summer solstice, resulting in slightly lower PRA SED in Melbourne for this time of day at summer solstice. However, solar noon occurred at a slightly later time-of-day in Melbourne than in Denver for the corresponding days to solstice (e.g., 13:33 21 February 2014, 13:18 21 December 2013 compared with 13:03 21 August 2013, 21 June 2013 at 13:01, respectively [39]). This accounts for the larger discrepancy in solar elevation angles in Melbourne between 11 am and 3 pm on a given date compared with the relatively equal solar elevation angles between 11 am and 3 pm in Denver for the corresponding dates. Overall, there were only a few small variations in SED apparent by city.

### 3.6. UV Exposure by Shade Design

As described earlier, the features of the built-shade varied by city. However, the majority of structures were tensioned shade sail designs (64% shade sails, 25% hip and ridge, and 11% cantilever). The shade structures built in Denver had on average a smaller cloth size than in Melbourne, ranging from 20.8 m^2^ to 111.5 m^2^ and from 83 m^2^ to 177 m^2^, respectively. The shade built in Denver also had lower entry heights than in Melbourne (Denver: 2.4–3.0 m; Melbourne: 3.5–3.6 m). Table 3 shows there was considerable variation in the UV levels under shaded PRAs during a 30 min potential exposure at posttest. Under clear skies the mean SED during posttest observations was $\bar{x}$ = 1.1 located at the shaded PRA center and $\bar{x}$ = 3.0 located at the PRA boundary. Lower SED were observed for the hip and ridge shade designs $\bar{x}$ = 0.6 SED, the medium sized shade structures $\bar{x}$ = 0.5 SED and for dark-colored shade cloth $\bar{x}$ = 0.9 SED. The highest SED under shade was observed for the light-colored cloth shade structures $\bar{x}$ = 1.4 SED. However, after adjusting for cloud, solar elevation and clustering within parks (Appendix B: Table A7, Table A8, Table A9, Table A10, Table A11, Table A12, Table A13, Table A14 and Table A15), there was no evidence that any of these variations by design type were statistically significant (indicated by the main effects Table A7, Table A10 and Table A13 in Appendix B).

## 4. Discussion

Shade construction produced a significant reduction in UV levels at shaded PRAs relative to the unshaded control PRAs. On average, the UV levels at the shaded PRAs were nearly three SEDs lower than the unshaded PRAs, during times of high solar elevation on cloudless days. Our findings demonstrate that relatively modest shade structures can provide a significant reduction in UV dose for a potential exposure of 30 min (equivalent to reducing UV by ~1.5 MED, a sunburning dose for a fair-skinned person with Fitzpatrick Skin Type I [4,7], the skin type at highest risk for skin cancer). However, during times of high solar elevation, the UV under shaded PRAs still approached two SED in 30 min, which could sunburn an individual with Fitzpatrick Skin Type I [4,7]. This implies that additional sun protection would be needed to avoid sunburn while using the shaded PRAs for longer than 30 min during peak UV in summer. The mean protection factors (PFs) for the posttest shade were relatively low (PF 3–17, $\bar{x}$
= 7.4) when measured against recommended shade protection for all day use of PF 15 or greater [33,42]. Nonetheless, individuals who used the shaded PRAs would have received a significantly lower dose of UV than in unshaded areas of the parks and could avoid a sunburn at least when outdoors for short periods and/or at lower UV levels.

Consistent with previous research, environmental conditions were a strong determinant of UV levels under the shade [32,33,43]. This variability in solar elevation (a function of latitude, time of day, and days to summer solstice), cloud cover and ozone makes it difficult to directly compare UV performance of shade structures across studies. A varied range of PFs are described in the few studies that measured UV in situ during summer for shade cloth structures with similar features of construction. For example, Gies and Mackay [33] measured PFs of 1.9 to 170.0 (direct UV) around solar noon under 14 shade cloth structures located in schools in New Zealand. Additionally, PFs of approximately two to 16 were recorded for shade structures over 16 children’s swimming pools in Victoria, Australia during peak UV hours, with shade size not a strong predictor of PF [38]. Relatively low PFs have also been observed for roofed-shade structures, for example three shade pavilions (15–32 m^2^) located in Queensland, Australia had PFs ranging from two to six, with size being a predictor of protection [43]. Tree shade, by contrast, also provided variable UV protection. The range of PFs measured for tree species included PF two to three for the oak and maple trees, PF two to six for gum (eucalyptus) trees, and PF three to 20 for the Australian she-oak trees, with some variation in PFs attributed to proximity to the trunk and cloud cover [23].

Several studies described shade design and locality features that affect UV levels under built-shade [33,38,43,44,45] but similar differences did not emerge in the current study. The non-random assignment of design features in this current study may explain the lack of an effect of them on SED for shaded PRAs at posttest. Most of the shade structures were built over areas with grass, soil, or wood chips, rather than paved areas, within the parks. For this reason, we expected low reflectivity of UV under the shade, given that the measured albedo (reflectance) of these surfaces is typically low (e.g., grass 4% versus concrete 10% [43,44,45]). Size of the shade will logically have an effect on UV, given it will directly impact the amount of sky view reduced [43]. However, there appears to be little evidence of a large effect of shade size on the measured UV levels of shade cloth structures in situ [33,38]. Possibly, other factors confounded the analysis, for example the UPF of the shade cloth or orientation to the sun’s path, or the variation in size of observed structures in these studies was too limited to detect differences. The shade structures in our study used maximum UPF shade cloth; therefore, UPF was not a confounding factor. Nonetheless, there is evidence that designs that have a sided-roof structure, such as hip and ridge styles, may have reduced sky view and offered greater protection from scattered and diffuse UV compared with more open-styles such as shade sails and cantilever designs [1,32,33]. Although not statistically significant in the current study, the measured UV and PFs for the different designs were in the expected directions.

Modern shade cloths can be manufactured with materials and coatings that ensure a high UPF for all colors [1]. The shade cloth used for structures in this study all had high UPF regardless of cloth color (Gale Pacific). The lighter shade cloth structures tended to have higher pole heights as well, which possibly bias any analysis of variation in UV by cloth color, given that the sun may reach further under higher shade structures. For better UV protection and thermal comfort, Poppinghaus [1] recommends use of white Teflon on the sun-side of fabric to minimize heat, and for small shade structures, a dark or black side towards the earth to avoid short wave solar reflection and to lower total heat flux. In cooler climates with high UV, light warm-shade is preferred to promote usage [2]. An observational study of daycare centers in Germany suggested that shade sails are a popular choice for shade provision [46]. Shade cloth with the highest possible UPF should be chosen, as low UPF shade cloth could result in unexpected sunburn among users.

Several studies also recommend enclosing shade with nearby vegetation or wall structures to block scattered UV and maximize UV protection under the shade [32,33]. A totally enclosed shade space, while offering greater UV protection, may pose issues for park users wishing to enjoy views of the park as they engage with their outdoor environment. Care is also needed to avoid introducing climbing points onto the shade structure when adding other nearby structures, which could result in injury to users, damage to shade structures or liability for owners.

Construction of built-shade requires considerable expertise [47], with consideration of materials, orientation to the sun, height and size and presence of underground services and above-ground obstructions. The shade structures in the current study were designed for passive recreation. Shade constructed for active areas will require additional considerations to ensure safe designs, such as positioning of poles to avoid fall or striking zones. Shade in beach and pool settings will need to address the high albedo of sand, water and paved surfaces [44,45], just as we had to engineer shade structures in Denver to support the weight of snow in winter. Architects and experienced shade manufacturers may provide innovative architectural solutions to these issues [1,2]. Additionally, latitude will be a large determinant of seasonal effects on UV levels and choices for shade provision. In lower latitudes, UV levels may be high year-round, while in higher latitudes, use of polycarbonate awnings, deciduous trees or removable shade-cloth may be more appropriate to deal with seasonally low UV levels.

An additional element necessary for design of good shade is an understanding of how people use shade, what designs are attractive, and for which activities and for whom. Our previous research on the effects of introducing built-shade in the school setting [48,49] and as part of park refurbishments [31,50] suggest shade cloth structures are sufficiently attractive to increase the use of areas under the shade. Careful selection of the location of built-shade is needed to maximize benefits. The current study focused on reducing intense UV exposures close to solar noon in PRAs where people are likely to linger for longer times (e.g., to eat a meal, engage in social interaction or watch a sports match). However, it may be beneficial to design shade to be provided at other times of the day dependent on patterns of use of an area or other community needs [3].

The findings of these studies measuring UV under shade have also highlighted that the PF standard is, for the main part, an imperfect measure, showing considerable change in UV protection score with changing solar elevation and cloud cover for a single shade structure. Further research is needed to better evaluate the UV protection for the comparison and improvement of shade designs. In terms of protecting skin and eyes, the proportion of UV ‘reduced’ is not as important as the ‘actual’ level of UV exposure that still occurs under the shade. As a minimum quality, shade should reduce the risk of sunburn for short exposures (30 min or less) during times of peak ambient UV in summer. A SED below two for an exposure of 30 min achieves this goal for individuals with Fitzpatrick’s Skin Type I [4,7]; it would be useful if more studies reported SED under shade. Large population surveys in the USA [51,52] and Australia [53] have consistently found lower odds of sunburn incidence among people who stayed mostly under shade during their activities. Similarly, personal dosimetry studies have found good reduction in UV exposures under shade; with, for example, 91% lower UVB exposures of students who used a shade sail area over concrete compared with ambient exposure whilst wearing wrist dosimeters on Texas schoolgrounds [45].

This study’s rigorous comparison of the UV levels between shaded and unshaded PRAs over time provides strong evidence of the expected levels of reduction in UV attributable to the use of built-shade. Nonetheless, the study included only public parks in two cities, which may somewhat limit the generalizability of the findings to shade built in parks in semi-arid and temperate areas with similar ambient UV levels. In other settings the ambient UV and albedo of paved rather than grassed surfaces (e.g., in school grounds and city squares) may substantially alter the expected reduction in UV levels of built-shade compared with study parks. Therefore, further research in these settings may be warranted. Moreover, the findings may not be generalizable to types of shade other than built-shade using high UPF shade cloth. The limitation of the non-randomized assignment of the shade design features has been previously discussed. Finally, an element of measurement error occurs in all studies and the handheld UV meters may have under-estimated UV levels as compared with scientific instruments [36].

## 5. Conclusions

Built-shade provides a rapidly created complement to tree shade for urban parks, especially in arid areas and newer locations that can offer substantial UV protection. The protection offered appears sufficient for skin cancer prevention, at least for short times outdoors. Shade-seeking behavior should be promoted and supported when educating the public about the need to utilize sun protection measures. Architects and shade construction companies can produce a variety of engaging designs for such structures [1,2,3,54]. A very basic design with tensioned shade cloth can offer good shade and substantial protection from UV for the park setting, indicating it is a good investment for inclusion in park designs by city planners and urban designers. Nonetheless, UV under shade was highly variable depending on solar elevation and cloud cover and remained sufficiently high to require additional sun protection measures on clear sky days in summer to prevent sunburn during long periods outdoors. As well as research to inform good design for UV protection, it would be beneficial to provide onsite information to educate shade users about how to effectively use shade to reduce the risk of sunburn.

## Figures and Tables

**Figure 1 ijerph-19-10583-f001:**
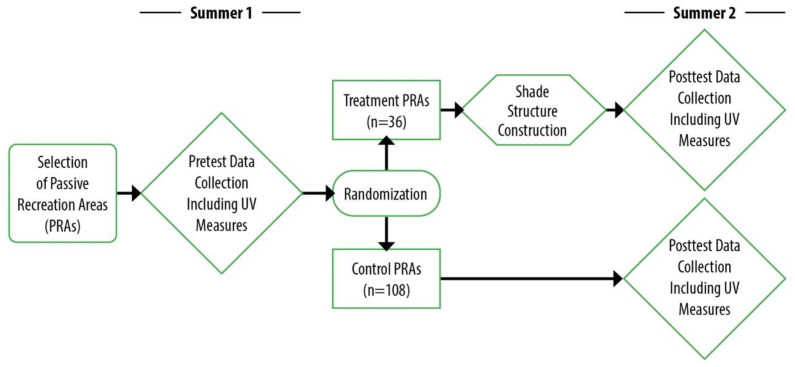
Schematic of workflow during two summers for each of three study waves, with the total number of PRAs for the study groups overall presented.

**Figure 2 ijerph-19-10583-f002:**
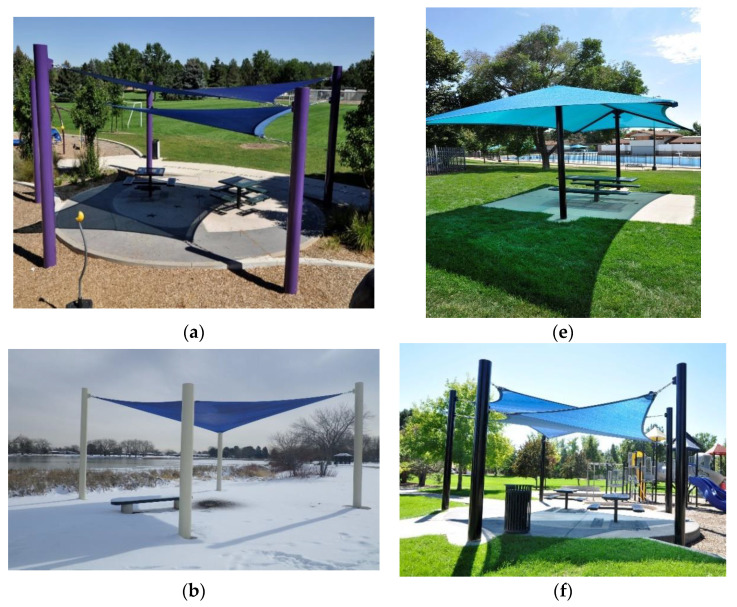
Shade structures built at study parks: (**a**) Double shade sails; (**b**) Single shade sail in winter snow; (**c**) Single shade sail during summer; (**d**) Hip and ridge design (**e**) Hip and ridge design; (**f**) Double shade sails; (**g**) Cantilever design; (**h**) Custom wrap shade design.

**Figure 3 ijerph-19-10583-f003:**
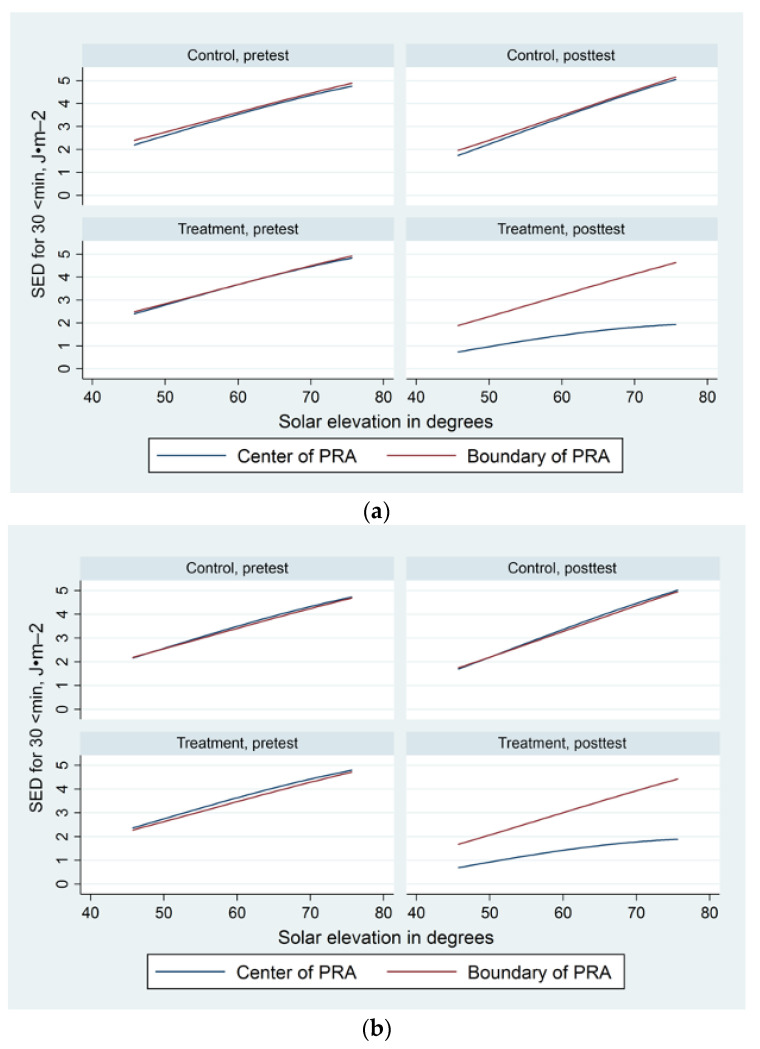
SED predicted: (**a**) at Denver PRAs by solar elevation during observations with clear skies; (**b**) at Melbourne PRAs by solar elevation during observations with clear skies.

**Figure 4 ijerph-19-10583-f004:**
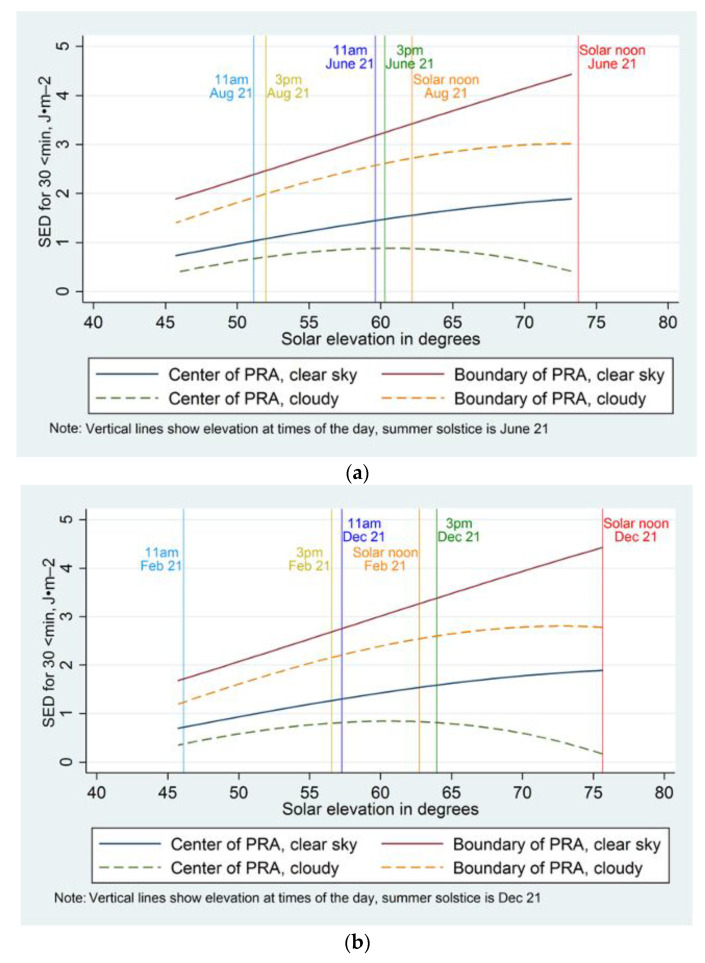
SED predicted: (**a**) at shaded PRAs in Denver during posttest by solar elevation and cloud; (**b**) at shaded PRAs in Melbourne during posttest by solar elevation and cloud.

**Table 1 ijerph-19-10583-t001:** Distribution of PRA observation conditions, percentages or means and (SD).

Measurement Conditions	Denver	Melbourne	Treatment	Control	Pretest	Posttest	Total
	N = 576	N = 568	N = 284	N = 860	N = 571	N = 573	N = 1144
Clear skies	41.5%	19.0%	34.2%	29.1%	28.6%	32.1%	30.3%
Cloud present	58.5%	81.0%	65.9%	70.9%	71.5%	67.9%	69.7%
Minutes to solar noon ^1^	−36.3 (69.0)	−47.1 (60.4)	−33.1 (67.6)	−44.5 (64.0)	−42.4 (63.8)	−41.0 (66.4)	−41.6 (65.1)
Days to summer solstice	38.1 (26.6)	40.0 (26.3)	39.5 (27.4)	38.9 (26.2)	28.6 (22.0)	49.4 (26.4)	39.0 (26.5)

^1^. Reporting minutes at the end of the PRA observation period when the UV readings were conducted c.f. the minutes to solar noon at the start of the PRA observation analyzed and reported in previous publications.

**Table 2 ijerph-19-10583-t002:** Unadjusted solar UV per 30 min at *n* = 144 PRAs by position in PRA and test period-mean SED, in units of J/m^2^, (SD).

Measurement		Pretest		Posttest	
Location	N	Treatment PRAs ^1^ (N = 36 PRAs)	Control PRAs (N = 108 PRAs)	Treatment PRAs ^2^ (N = 36 PRAs)	Control PRAs (N = 108 PRAs)
PRA Center	1144	3.39 (1.32)	3.21 (1.38)	0.93 (1.18)	2.85 (1.36)
PRA Boundary	1144	3.34 (1.35)	3.22 (1.38)	2.59 (1.46)	2.86 (1.34)

^1^. Before shade construction; ^2^. After shade construction.

**Table 3 ijerph-19-10583-t003:** UV exposure (SED/sun protection factor (SPF)) during 30 min under shaded PRAs, posttest by shade design, city and cloud cover.

PRA Shade Design Type		Denver		Melbourne			Overall	
		Clear Sky	Cloud Present	Clear Sky	Cloud Present			
	No. Obs.	PRA Center	PRA Center	PRA Center	PRA Center	PRA Center	PRA Boundary	PRA Center
		SED Mean (SD)	SED Mean (SD)	SED Mean (SD)	SED Mean (SD)	SED Mean (SD)	SED Mean (SD)	SPF (SD)
Overall shaded PRAs	142	0.9 (1.1)	0.5 (0.5)	1.4 (1.7)	1.1 (1.3)	0.9 (1.2)	2.6 (1.5)	7.4 (14.4)
Shade design								
Cantilever Hip and ridge ^a^	15 35	1.2 (0.8) 0.9 (1.4)	0.5 (0.1) 0.3 (0.3)	-- --	-- --	1.0 (0.8) 0.6 (1.0)	2.8 (0.9) 3.0 (1.0)	4.2 (2.0) 15.6 (25.3)
Shade sail	92	0.7 (0.9)	0.8 (0.9)	1.4 (1.7)	1.1 (1.3)	1.1 (1.3)	2.4 (1.6)	4.7 (6.6)
Shade cloth color								
Dark	122	0.9 (1.1)	0.5 (0.5)	1.2 (1.7)	1.0 (1.2)	0.9 (1.1)	2.7 (1.4)	8.2 (15.4)
Light	20	--	--	2.2 (1.7)	1.2 (1.4)	1.4 (1.5)	2.2 (1.7)	2.7 (2.8)
Shade cloth size, m^2^								
20–35	42	1.1 (1.1)	0.5 (0.6)	--	--	0.7 (0.9)	3.1 (1.1)	9.4 (6.9)
35–80	24	0.6 (0.8)	0.4 (0.4)	--	--	0.5 (0.6)	3.2 (1.1)	17.2 (32.1)
≥81	76	1.6 (2.3)	1.5 ^b^	1.4 (1.7)	1.1 (1.3)	1.2 (1.4)	2.1 (1.6)	3.3 (3.1)
Shade cloth entry height, m								
<3.0	20	0.9 (1.4)	0.5 (0.7)	--	--	0.6 (0.8)	3.2 (1.0)	16.7 (33.3)
≥3.0	122	0.9 (1.1)	0.4 (0.4)	1.4 (1.7)	1.1 (1.3)	1.0 (1.2)	2.5 (1.5)	5.8 (7.0)

The UV measurements after shade construction at 36 PRAs, comprised 144 center and 144 boundary SEDs measurements, including *n* = 2 with missing cloud observations; Standard Erythemal Dose (SED) = ((measured UVI units at PRA × 0.025 W/m^2^) × number of seconds exposure i.e., 30 × 60)/100 J/m^2^ per SED); Sun Protection Factor = Boundary PRA SED/Center PRA SED; ^a^
*n* = 1 custom wrap shade design was analyzed as a hip and ridge structure; ^b^
*n* = 1 observation; -- no relevant observation for this table cell.

## Data Availability

The datasets generated and/or analysed during the current study are available in the openICPSR repository. [http://doi.org/10.3886/E177661V1, accessed on 10 August 2022]. The trial registration is Clinicaltrials.gov identifier NCT02971709.

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
