# Peer review of "Solar UV Measured under Built-Shade in Public Parks: Findings from a Randomized Trial in Denver and Melbourne"

_ijerph, 2022, doi:10.3390/ijerph191710583_

Round 1
Reviewer 1 Report
Summary
The paper 'Solar UV measured under built-shade in public parks: Findings from a randomized trial in Denver and Melbourne' by Dobbinson et al. is well written and organized. Methodology seems to be adequate to the aim of the research and is mostly clearly described. Results are provided in the form of understandable tables and graphs, although axis/column labeling could be improved at some points (see below for details). The work aims to provide information on the UV protection provided by different types of built-shade and to examine environmental predictors of erythemal dose under these built-shade structures. By presenting results from a randomized controlled trial that show that a substantial reduction in exposure to UV can be achieved, this research can e.g. help decision makers in towns etc. to install an evidence-based method for sun protection in public parks.
General comments:
In this manuscript, the separation between the measurand "erythemal dose" and the associated unit "standard erythema dose" is not sharp. There is a publication by the International Commission on Illumination (ISO/CIE 17166:2019) that explains these terms in detail. One cannot say that the unit of a SED is 100 J/m² (as the authors do in line 191), but 1 SED is identical to 100 J/m² weighted with the CIE erythema reference action spectrum (the differences between the CIE action spectrum and the Diffey action spectrum used in the Solarmeters to measure the UV index are small and well within the uncertainty of the biological data on which the action spectrum was based, as shown in [Webb, A.R., Slaper, H., Koepke, P. and Schmalwieser, A.W. (2011), Know Your Standard: Clarifying the CIE Erythema Action Spectrum. Photochemistry and Photobiology, 87: 483-486. https://doi.org/10.1111/j.1751-1097.2010.00871.x].
All over the manuscript, the authors speak of having measured and analyzed SED or “UV levels” or “mean UV”. I think it’s important to be precise here and replace these terms with “erythemal dose” except for those cases when SED is really used as the unit of erythemal dose.
The authors should also adjust labels and captions of graphs and tables accordingly. For example, in figures 2 and 3, the y axis is labeled as “SED for 30 <min, J.m-2”. This could be changed to “erythemal dose for an exposure duration of 30 min, given in SED”, with the caption of the figure explaining that 1 SED = 100J/m² weighted with the CIE erythema reference action spectrum.
Detailed comments
Line 43: although the sun may be very high in summer in Denver or Melbourne, the sun cannot be directly overhead as both cities‘ latitude is not between the Northern and Southern Tropic
Line 88-94: I think this belongs to the methods section, not the introduction
Line 94-96: The secondary aim described here is the same as the aim described in lines 80-82
Figure 1c: The number „616-11“ in the photo doesn’t provide any information to the reader and might be confusing. Maybe the authors could remove it.
Line 191-193: I think this sentence can be somewhat misleading. A minimal erythemal dose is the erythemal dose sufficient to induce erythema in the corresponding skin type. I think it’s not clear here that the value of 1 MED is 2 SEDs (the exact value varies somewhat between sources) only for skin type 1, but is higher for the more melano-competent skin types 2-6.
Table 1: There are some differences in the values of minutes to solar noon reported here and in previous publications of the authors ([x] = number of citation in authors‘ reference list). Those differences are too big to be explained by the small variation in Ns between the analyses:
|
|
[35] (N=580) |
[31] (N=576) |
Current manuscript (N=571 pretest, N=573 posttest) |
|
Minutes to solar noon, mean (sd) |
|
|
|
|
Pretest |
55 (35) |
84 (49) |
-42.4 (63.8) |
|
Posttest |
NA (publication only includes pretest measurements) |
82 (52) |
-41.0 (66.4) |
Line 394: As described above, the term MED is not limited to a certain skin type. It should also generally not be used as a unit but only as a threshold value (ISO/CIE 17166:2019). Thus, I would not recommend saying “equivalent to reducing UV by ~1.5 MED, a sunburning dose for a fair-skinned person”. The authors could replace this with “a difference in erythemal dose of 3 SED exceeds the MED of skin type 1 by a factor of 1.5”.
Author Response
Reviewer 1
|
Reviewer’s Comment |
Authors’ Response |
||||||||||||||||
|
Summary The paper 'Solar UV measured under built-shade in public parks: Findings from a randomized trial in Denver and Melbourne' by Dobbinson et al. is well written and organized. Methodology seems to be adequate to the aim of the research and is mostly clearly described. Results are provided in the form of understandable tables and graphs, although axis/column labeling could be improved at some points (see below for details). The work aims to provide information on the UV protection provided by different types of built-shade and to examine environmental predictors of erythemal dose under these built-shade structures. By presenting results from a randomized controlled trial that show that a substantial reduction in exposure to UV can be achieved, this research can e.g. help decision makers in towns etc. to install an evidence-based method for sun protection in public parks. |
|
||||||||||||||||
|
General comments: In this manuscript, the separation between the measurand "erythemal dose" and the associated unit "standard erythema dose" is not sharp. There is a publication by the International Commission on Illumination (ISO/CIE 17166:2019) that explains these terms in detail. One cannot say that the unit of a SED is 100 J/m² (as the authors do in line 191), but 1 SED is identical to 100 J/m² weighted with the CIE erythema reference action spectrum (the differences between the CIE action spectrum and the Diffey action spectrum used in the Solarmeters to measure the UV index are small and well within the uncertainty of the biological data on which the action spectrum was based, as shown in [Webb, A.R., Slaper, H., Koepke, P. and Schmalwieser, A.W. (2011), Know Your Standard: Clarifying the CIE Erythema Action Spectrum. Photochemistry and Photobiology, 87: 483-486. https://doi.org/10.1111/j.1751-1097.2010.00871.x]. All over the manuscript, the authors speak of having measured and analyzed SED or “UV levels” or “mean UV”. I think it’s important to be precise here and replace these terms with “erythemal dose” except for those cases when SED is really used as the unit of erythemal dose. The authors should also adjust labels and captions of graphs and tables accordingly. For example, in figures 2 and 3, the y axis is labeled as “SED for 30 <min, J.m-2”. This could be changed to “erythemal dose for an exposure duration of 30 min, given in SED”, with the caption of the figure explaining that 1 SED = 100J/m² weighted with the CIE erythema reference action spectrum. |
We acknowledge that the UV data measured by the Solarmeters does not provide a highly precise measure of SED. However, ‘erythemal dose’ is not a recognised unit and the reviewer indicates that the small discrepancy, between our measure and the standard utilising the CIE weighted spectra, is not likely to be biologically meaningful.
We have addressed the Reviewer’s concerns about precision by providing justification for using the SED measure for our Solarmeters data in the Statistical Analysis section, citing Webb (Lines 250-258).
|
||||||||||||||||
|
Line 43: although the sun may be very high in summer in Denver or Melbourne, the sun cannot be directly overhead as both cities‘ latitude is not between the Northern and Southern Tropic |
The manuscript was revised accordingly, using the phrase “when the sun is at its seasonally highest for the locality” (Line 48). |
||||||||||||||||
|
Line 88-94: I think this belongs to the methods section, not the introduction |
We agree and have merged this text into paragraph 1 of the Methods section (Lines 95-110). |
||||||||||||||||
|
Line 94-96: The secondary aim described here is the same as the aim described in lines 80-82
|
The original sentence was eliminated and the sentence originally on lines 94 to 96 have been moved to the methods section (Lines 103-105) where the authors believe a description of the more specific secondary aim of this study is appropriate. |
||||||||||||||||
|
Figure 1c: The number „616-11“ in the photo doesn’t provide any information to the reader and might be confusing. Maybe the authors could remove it. |
This study site label was removed from the photo in Figure 1c. |
||||||||||||||||
|
Line 191-193: I think this sentence can be somewhat misleading. A minimal erythemal dose is the erythemal dose sufficient to induce erythema in the corresponding skin type. I think it’s not clear here that the value of 1 MED is 2 SEDs (the exact value varies somewhat between sources) only for skin type 1, but is higher for the more melano-competent skin types 2-6. Line 394: As described above, the term MED is not limited to a certain skin type. It should also generally not be used as a unit but only as a threshold value (ISO/CIE 17166:2019). Thus, I would not recommend saying “equivalent to reducing UV by ~1.5 MED, a sunburning dose for a fair-skinned person”. The authors could replace this with “a difference in erythemal dose of 3 SED exceeds the MED of skin type 1 by a factor of 1.5”. |
We acknowledge that MED is not restricted to a certain skin type. We revised the text in the Statistical Analysis section (Lines 250-258) to provide the rationale for calculating MED specifically for the Fitzpatrick Skin Type I group rather than for ‘fair skin’, the previously less precise term specified. This change was also made throughout the manuscript including at the beginning of the discussion.
|
||||||||||||||||
|
Table 1: There are some differences in the values of minutes to solar noon reported here and in previous publications of the authors ([x] = number of citation in authors‘ reference list). Those differences are too big to be explained by the small variation in Ns between the analyses:
|
We thank the reviewer for their due diligence in comparing the current study with previous publications for the study. We reviewed the apparent discrepancies for the two most comparable papers (Table 1 in the current paper and the values in the Am J Public Health paper), i.e. those analysing pre-test and post-test data for ’minutes to solar noon.’
There are four variables in the analysis dataset for ‘minutes to solar noon’ depending on whether the start or end of the observation period and whether we use the numeric difference or the absolute value of the difference. Also, there is the full sample (n=1152) or the SEDs analysis sample (n=1144). In the IJERPH paper we use the SEDs analysis sample and numeric difference and the end of the observation period, whereas in Am J Public Health we use the absolute value of the numeric difference, the start of the observation period, and the full sample.
We believe that the variables and samples used were appropriate for the analyses in these two papers. The start of the observation period was most appropriate for the analysis of use of the PRA (reported in the Am J Public Health paper), while the end of the observation period at the PRA was when the UV readings were conducted and are thus appropriate for the current analysis of SED. We added a table note to Table 1 to clarify this difference from the previous publications. |
Reviewer 2 Report
The study by Dobbinson et al. describes findings from a randomized trial on the association between solar UV measurements and various predictors related to built-shade structures in public parks.
The manuscript is well written and the results are graphically clearly presented. Overall, the study provides an extensive assessment on important factors that can or should be considered in designing outdoor public recreational areas with shade structures.
I have only some very minor comments and suggestions:
- In Section 3.2., there is one subsection 3.2.1, but no other subsections as 3.2.2, 3.2.3…. Maybe the sections and subsections were mixed up here?
- In line 275, I would suggest to change the term „times“ to „units“ or to rephrase the sentence. Since the coefficients from a linear regression are not multiplicative, the term „(four) times“ is misleading.
- In line 287: „there was no significant difference…“, however, the corresponding 95% CI in line 289 does not contain the 0 value („-2.32 to -0.76“) and therefore indicates a significant difference? (s. also Table A3)
- In Table B2: the correct p-value for „1-Cantilever“ is 0.0084 (instead of 0.084)
Author Response
Reviewer 2
|
Reviewer comment |
Authors’ response |
|
The study by Dobbinson et al. describes findings from a randomized trial on the association between solar UV measurements and various predictors related to built-shade structures in public parks. The manuscript is well written and the results are graphically clearly presented. Overall, the study provides an extensive assessment on important factors that can or should be considered in designing outdoor public recreational areas with shade structures. |
|
|
- In Section 3.2., there is one subsection 3.2.1, but no other subsections as 3.2.2, 3.2.3…. Maybe the sections and subsections were mixed up here? |
We have merged the sub-section text into the main paragraph (Lines 325-326) to remove this potential confusion.
|
|
- In line 275, I would suggest to change the term „times“ to „units“ or to rephrase the sentence. Since the coefficients from a linear regression are not multiplicative, the term „(four) times“ is misleading. |
We agree that a coefficient from a linear regression should not imply multiplication so we revised the text to use the terms “units” (see Line 348). |
|
- In line 287: „there was no significant difference…“, however, the corresponding 95% CI in line 289 does not contain the 0 value („-2.32 to -0.76“) and therefore indicates a significant difference? (s. also Table A3) |
Thank you for identifying this error. The sign of the upper 95% CI was incorrect. It actually is -0.76 not +0.76 (Table A3 and Line 362). |
|
- In Table B2: the correct p-value for „1-Cantilever“ is 0.0084 (instead of 0.084) |
The Cantilever p-value in Table B2 has been corrected to 0.0084 (written at 0.008 to three decimal places in the table). |

Reviewer 3 Report
The manuscript reports results from a randomized controlled trial comparing measured UV levels between passive recreation areas (PRAs) with built-shade and unshaded. The study has been performed over summer months in 144 public parks in two cities. The topic is important, the methodology is sound, the presentation of results is very detailed and complex, the discussion is comprehensive.
Specific remarks:
Abstract:
(i) The sample size of the study is n=144 (and not 1144) since there are 144 experimental units (each contributing several measurements). Therefore, l. 17-20 has to be rephrased.
(ii) Please add the information when the study has been performed. The information that the study has been conducted long ago should be given in the abstract.
(iii) When providing the numerical estimate (-3.47) describing the difference between shaded and unshaded PRAs, it should be added that this is the adjusted estimate from a model including covariates (to be named).
Statistical Analysis:
l. 191: MEDs vary strongly between individuals, primarily depending on skin type. While SEDs have a clear meaning, the "translation" of SED into MED is more complex. Different classifications of SEDs dependent on skin type into MEDs have been proposed. Equaling an SED of 2 with the MED for fair-skinned individuals is one - but not the only one - of the proposals. I wonder why MEDs are introduced at all. All data and results are given in terms of SEDs. After the introduction of the MED in l. 191, the term MED appears only once in the discussion (where it could be easily avoided by rephrasing the sentence).
l. 206: Please explain what is meant by "robust standard errors". Did you trim the data?
Results:
l. 253: The subsection 3.2 has only one subsubsection. I do not see the point of having the subsubsection 3.2.1 if there is no additional 3.2.2.
Table 1: Check the total column for "Days to summer solstice", the number has to be typo.
l. 322: The minus in front of the lower bound of the CI (i.e. -18.89) should not be written in a separate line.
Table 3: A comparative evaluation of the effect of shade design on UV levels would, of course, be of high interest. The present study is, however, not designed to give a methodologically sound answer to this question. The authors are aware of the problem of non-randomized comparisons and have tried to address it in a series of statistical models incorporating potential confounders and effect-modifiers. The (inconclusive) results of these sophisticated analyses are given in Appendix B, while in the main text only unadjusted results are presented in Table 3. To my opinion, this gives the unadjusted data more weight than they deserve. The authors should present adjusted results in section 3.6 and shift Table 3 to the appendix. Of course, it is challenging to find a way to condense the complex results from modeling, but the current version to show only the "simple" data is misleading.
Discussion
A really excellent and comprehensive discussion of the many aspects this study! The only thing to add would be some more words about investigations addressing shade in other settings. While this study considered the setting "public park" and touched upon the school setting, shade in daycare centers where young children are supervised represents a topic worth mentioning. I remember a study in IJERPH by Fiessler et al. addressing the issue (see doi:10.3390/ijerph15091793).
Author Response
Reviewer 3
|
Reviewer comment |
Authors’ response |
|
The manuscript reports results from a randomized controlled trial comparing measured UV levels between passive recreation areas (PRAs) with built-shade and unshaded. The study has been performed over summer months in 144 public parks in two cities. The topic is important, the methodology is sound, the presentation of results is very detailed and complex, the discussion is comprehensive. |
|
|
Abstract: (i) The sample size of the study is n=144 (and not 1144) since there are 144 experimental units (each contributing several measurements). Therefore, l. 17-20 has to be rephrased. |
The abstract text was revised to clarify the experimental units (Line 20). |
|
(ii) Please add the information when the study has been performed. The information that the study has been conducted long ago should be given in the abstract. |
This information was added to the abstract on line 21.
|
|
(iii) When providing the numerical estimate (-3.47) describing the difference between shaded and unshaded PRAs, it should be added that this is the adjusted estimate from a model including covariates (to be named). |
The details of the covariates were added to the abstract on lines 25-26 in the sentence reporting the decrease in UV under shade. |
|
Statistical Analysis: l. 191: MEDs vary strongly between individuals, primarily depending on skin type. While SEDs have a clear meaning, the "translation" of SED into MED is more complex. Different classifications of SEDs dependent on skin type into MEDs have been proposed. Equaling an SED of 2 with the MED for fair-skinned individuals is one - but not the only one - of the proposals. I wonder why MEDs are introduced at all. All data and results are given in terms of SEDs. After the introduction of the MED in l. 191, the term MED appears only once in the discussion (where it could be easily avoided by rephrasing the sentence). |
The concept of MED is an important one for determining risk of sunburn. The number of times an individual is sunburnt is a significant risk factor for melanoma. Prevention advocates recommend that people reduce their UV exposure and avoid sunburn. It is therefore a useful biological marker of excessive UV exposure. Thus, we have retained the sentence on MED, but specified that 2 SED are equivalent to 1 MED for Fitzpatrick Skin Type I. We also explain why MED is relevant for the interpretation of the study findings. See lines 250-258.
|
|
l. 206: Please explain what is meant by "robust standard errors". Did you trim the data?
|
The data were not trimmed. Robust standard errors refer to the Huber/White/sandwich estimator used to calculate the standard errors. The calculation gives valid inference in the situation where the residuals are independent of the covariates, but not necessarily identically distributed. In our case there was some evidence that the variance of the residuals varied with solar elevation, so robust standard errors were used. We revised the manuscript text to include a brief definition of robust standard errors and why we used them in the current analysis (Lines 271-275).
For further information on Robust Standard Errors please refer to the Stata Manual and the following references: · Huber, P. J. 1967. The behavior of maximum likelihood estimates under nonstandard conditions. In Vol. 1 of Proceedings of the Fifth Berkeley Symposium on Mathematical Statistics and Probability, 221–233. Berkeley: University of California Press. · White, H. L., Jr. 1980. A heteroskedasticity-consistent covariance matrix estimator and a direct test for heteroskedasticity. Econometrica 48: 817–838. · White, H. L. 1982. Maximum likelihood estmation of misspecified models. Econometrica 50: 1–25. |
|
Results: l. 253: The subsection 3.2 has only one subsubsection. I do not see the point of having the subsubsection 3.2.1 if there is no additional 3.2.2. |
As noted above, we have merged the sub-section text into the main paragraph (Lines 225-226) to remove this potential confusion.
|
|
Table 1: Check the total column for "Days to summer solstice", the number has to be typo. |
The correct figures (mean: 39.0, SD: 26.5) have been added to the total column for “Days to summer solstice” in Table 1. |
|
l. 322: The minus in front of the lower bound of the CI (i.e. -18.89) should not be written in a separate line.-
|
This occurred due to the journal’s paragraph style auto formatting. We corrected the formatting to close up the sign to the numerical value (Line 354). Following revisions there was a similar issue on line 412. We added a manual line break (using layout, breaks, text wrapping) and have checked that there are no other similar formatting errors in the final manuscript. |
|
Table 3: A comparative evaluation of the effect of shade design on UV levels would, of course, be of high interest. The present study is, however, not designed to give a methodologically sound answer to this question. The authors are aware of the problem of non-randomized comparisons and have tried to address it in a series of statistical models incorporating potential confounders and effect-modifiers. The (inconclusive) results of these sophisticated analyses are given in Appendix B, while in the main text only unadjusted results are presented in Table 3. To my opinion, this gives the unadjusted data more weight than they deserve. The authors should present adjusted results in section 3.6 and shift Table 3 to the appendix. Of course, it is challenging to find a way to condense the complex results from modeling, but the current version to show only the "simple" data is misleading. |
The authors believe that including the details of the higher-level models analysing the effect of shade design on UV levels would overwhelm the paper. The unadjusted results in Table 3 are useful to assist readers to better understand the analyses and findings. In the discussion we have placed these findings in the context of other studies of the effects of UV (Lines 521-539) and it is useful to note that direction of effects for various features were in the expected directions, based on data from these observational studies. We have also highlighted in three sections of the manuscript that the results on shade design effects are limited by the non-random assignment of these design features (Lines 296, 523, and 623). |
|
Discussion A really excellent and comprehensive discussion of the many aspects this study! The only thing to add would be some more words about investigations addressing shade in other settings. While this study considered the setting "public park" and touched upon the school setting, shade in daycare centers where young children are supervised represents a topic worth mentioning. I remember a study in IJERPH by Fiessler et al. addressing the issue (see doi:10.3390/ijerph15091793). |
We agree that provision of shade in settings such as daycare centers is important. The study by Fiessler et al. describes the extent and type of shade observed in 243 daycare centers in Germany. That study reveals the popularity of shade cloth structures as a choice for shade provision. As such it is a useful addition to the discussion (Lines 548-550). |

Reviewer 4 Report
Dear authors,
I thank you for this manuscript and congratulate you on this work. Overall, I find this work interesting, although I would like to make a few points.
Especially the sections on material and methods, as well as the results, I find difficult to read. Maybe it's just me, but I didn't really find access to these sections. I can't quite put my finger on it either, but there is far too much information added in brackets, and many of the sentences and thoughts are simply "unrythmic" to read. There is also too much assumed knowledge to possibly explain to people outside the subject. I would recommend reorganising much of this if the other reviewers feel the same way.
- In the introduction, please focus mainly on NMSC.
- Please include the geographical coordinates of Denver and Melbourne.
- 2.2 I would support with an illustration to workflow, perhaps this can be done elsewhere to make chapters 2 and 3 more readable.
- Maybe I missed it, but how was the height of the PRAs considered and the height measurement from the ground and the distance of the measurement from the PRA?
- Why was the measurement made handheld and not on a tripod?
- How was the calibration of the measuring instruments ensured/performed.
- Why was the wind measurement not carried out with a simple measuring device?
- When it comes to skin light types, please quote Fitzpatrick.
Now these were the points that stood out to me. Thanks again for letting me read the manuscript!
Author Response
Reviewer 4
|
Reviewer comment |
Authors’ response |
|
Dear authors, I thank you for this manuscript and congratulate you on this work. Overall, I find this work interesting, although I would like to make a few points. |
|
|
Especially the sections on material and methods, as well as the results, I find difficult to read. Maybe it's just me, but I didn't really find access to these sections. I can't quite put my finger on it either, but there is far too much information added in brackets, and many of the sentences and thoughts are simply "unrythmic" to read. There is also too much assumed knowledge to possibly explain to people outside the subject. I would recommend reorganising much of this if the other reviewers feel the same way. |
We agree that the flow of the methods section needed to be improved and have made revisions throughout. We have also removed any unnecessary brackets where appropriate. At the same time, we have addressed critiques of the methods from other reviewers and these revisions should also improve the clarity of these sections. Some aspects of UV science are very technical and we have minimised this highly technical language where possible. We also added a flow chart (see Figure 1) to clarify the study procedures to those people less familiar with randomised trials. |
|
- In the introduction, please focus mainly on NMSC.
|
We added text in the Introduction that non-melanoma skin cancer is more prevalent than melanoma in both Australia and the USA and that they are both caused by UV exposure (Lines 58-60). |
|
- Please include the geographical coordinates of Denver and Melbourne. |
The geographical coordinates were added to the methods section (lines 100-101). |
|
- 2.2 I would support with an illustration to workflow, perhaps this can be done elsewhere to make chapters 2 and 3 more readable. |
We have added a workflow illustration to Materials and Methods (Figure 1, positioned at line 111). |
|
- Maybe I missed it, but how was the height of the PRAs considered and the height measurement from the ground and the distance of the measurement from the PRA? |
We added these details to 2.2.1. (Lines 169-170). |
|
- Why was the measurement made handheld and not on a tripod? |
We added the rationale for the use of the Solarmeter by hand to text in the methods where the reliability of the UV readings is described (Lines 212-215). |
|
- How was the calibration of the measuring instruments ensured/performed.
|
The calibration of the measuring instruments was done by the manufacturer. Our pilot study indicated the findings were relatively reliable as compared with the UV science station measures. We described this on Line 206 and 216-219, respectively. |
|
- Why was the wind measurement not carried out with a simple measuring device?
|
Behavioural scientists have developed measures of wind relevant to human perception. For the purpose of this study these measures were deemed adequate relative to the expense and logistical difficulties of carrying around large equipment to the 144 parks on multiple observation dates. Additionally, we wished for the research staff to be unobtrusive and felt such equipment would call attention to them. We note this on Line 239. |
|
- When it comes to skin light types, please quote Fitzpatrick. |
We have now specified the relevant Fitzpatrick Skin Types for all references to light skin types throughout the manuscript. |

Round 2
Reviewer 3 Report
The authors addressed most of my - and the other reviewers' - remarks appropriately. The Methods section of the revised version describes now more clearly what has been done (in the study and in the analysis).
The only point of concern relates to the presentation of results on the effect of shade design on SEDC30. The authors insist on presenting raw (unadjusted) data on SEDC30 in different subgroups defined by shade design in the main text (subsection 3.6 and Table 3), while showing more detailed results from modeling SEDC30 in a number of different models incorporating covariates (and partly restricting the data set) only in the Appendix B. What puzzles me most is the fact that the factor shade design is operationalized differently in Table 3 and in Appendix B. In the results shown in Appenix B three types of shade design are addressed, while in Table 3 two of them were merged. In particular, the Cantilever design (which showed the strongest effect on SEDC30 in the different models) was combined with the hip and ridge design (which showed the weakest effect in the different models). By doing so, Table 3 gives the impression that differences between shade designs with respect to SEDC30 are only modest. The authors do not give a reason for merging different shade designs and thus should revise their Table 3.
Author Response
Reviewer 3
|
Reviewer’s Comment |
Authors’ Response |
|
The authors addressed most of my - and the other reviewers' - remarks appropriately. The Methods section of the revised version describes now more clearly what has been done (in the study and in the analysis). |
|
|
The only point of concern relates to the presentation of results on the effect of shade design on SEDC30. The authors insist on presenting raw (unadjusted) data on SEDC30 in different subgroups defined by shade design in the main text (subsection 3.6 and Table 3), while showing more detailed results from modeling SEDC30 in a number of different models incorporating covariates (and partly restricting the data set) only in the Appendix B. What puzzles me most is the fact that the factor shade design is operationalized differently in Table 3 and in Appendix B. In the results shown in Appendix B three types of shade design are addressed, while in Table 3 two of them were merged. In particular, the Cantilever design (which showed the strongest effect on SEDC30 in the different models) was combined with the hip and ridge design (which showed the weakest effect in the different models). By doing so, Table 3 gives the impression that differences between shade designs with respect to SEDC30 are only modest. The authors do not give a reason for merging different shade designs and thus should revise their Table 3. |
We agree that the design type rows in Table 3 should be consistent with the models in Appendix B. We adjusted the analysis displayed in Table 3 to separate the Cantilever and Hip and Ridge design so that Table 3 is now consistent with the models in Appendix B. Please note that we also corrected an error that misclassified one of the shade structures as a shade sail when it was actually a hip and ridge design. Thus, the shade sail entries in Table 3 adjusted slightly. This re-analysis required making a small change in the text on line 399 but it did not change the results overall. |